# Tracing the Scientific Trajectory of Volunteered Cartography: The Case of OpenStreetMap

**Roberto Pizzolotto** 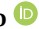

Department of Biology, Ecology and Earth Science, Università della Calabria, Ponte Bucci 4b,
87036 Rende, CS, Italy; roberto.pizzolotto@unical.it

**Abstract:** Where the streets have no name is probably the preferred place for a volunteer Open-StreetMapper. Launched in 2004, the Open Street Map project aimed to share geographical data based on volunteer mapping and led to the collection of geographical data from almost every country in the world within fifteen years. The increased dissemination of cartographic data via the Internet has been helpful in real life, socially, and has resulted in the number of published documents increasing rapidly. To evaluate the impact of volunteered cartography on scientific research, a science mapping approach was applied to the published literature on the Open Street Map project on the basis of co-occurrence and co-citation analyses, which showed that the main themes (conceptual network) were of technical relevance, collaboration among scholars and among institutes (social network) was not strong, and knowledge and ideas circulated within a limited network. In this study, documents published by OpenStreetMappers were analysed for the first time; thus, it was possible to highlight gaps in volunteered cartography and to discuss further improvements to the Open Street Map project.

**Keywords:** Open Street Map; bibliometry; social structure; applied cartography; scientific impact

## 1. Introduction

When the OpenStreetMap project began in 2004, it was probably unexpected that such fast faithful support by volunteer cartographers to cover the globe gathering geographic data would follow.

OpenStreetMap (hereinafter OSM) is a project founded in 2004 with the intention of sharing geographical data through volunteer mapping in order to produce cartographic products including mainly maps [1].

In 2009, five years after the project's inception, data gathering started to spread, likely from two diffusion centres of volunteer mappers from Europe and North America (see Figure S1), but commitment to the project was already spreading around the world. After ten years, nearly all the continents and islands of the world were mapped within the OSM database, with significant, and understandable, gaps in Greenland and nearby Arctic regions (i.e., North of Canada and Siberia), the Sahara Desert, and the core of Amazonas. After more than fifteen years, a huge amount of data were cartographed voluntarily (see Figure S2), even if the main gaps at the northern latitudes still remain. The cumulative sum of the volunteered geographic information [2] focused mainly on Europe, with several hot spots in Nepal, Indonesia, Japan, the Republic of Congo Central Africa, California and the USA West Coast. Those hot spots, apart from Europe and the USA, are points affected by environmental or health emergencies, e.g., earthquakes in Nepal and Japan and Ebola in Africa, which have been mapped thanks to the application of OSM data to tackle humanitarian issues by the Humanitarian OpenStreetMap Team initiative (HOT; https://www.hotosm.org/).

Volunteered cartography found an ideal place in OSM, because it is based on the contribution of a growing number of professional, as well as beginner, cartographers that gather geographical data and make them available under the open source philosophy.

The terrain reality is translated into maps by visual interpretation of freely accessible satellite images, field surveys and bulk import; therefore, the accuracy and reliability depend on image resolution and its time of acquisition, as well as the skilfulness of the cartographer (see [3] for a review). Then, the consistency of the data introduced can be validated by other voluntary cartographers or by direct verification on the ground by those who first introduced it. In the first years in which OSM gathered geographic data from nearly all land around the world, there was a greater concentration in highly industrialised countries, maybe as a consequence of better communication or collaboration among scholars, or both.

Given the successful spread of the OSM initiative, it is to be expected that it is linked to regular publications in scientific journals; therefore, the aim of the present paper is to evaluate the scientific trajectory, in terms of published papers, of volunteered cartography within OSM to determine if the result of OSM activities is mainly bounded to geographic data storing work (then, few generic documents are expected) or if it is accompanied by consistent document publishing activity. The main questions are: On the basis of bibliometric data: (i) how many and what kind of papers were published?; (ii) Are there preferred themes in publications?; (iii) Is it possible to find a network of scholars collaborating on OSM data?

## 2. Materials and Methods

To follow the hypothesis that the OSM initiative is linked to regular publication in scientific journals means that published documents mirror the impact on the scientific community of the geographical data recorded by OSM. Thus, bibliometric analysis is a useful tool to evaluate the scientific trajectory of volunteered cartography within the OSM initiative, because it is based on bibliographic records. A bibliographic record is generated for each published document as a set of information stored in several fields to describe a document with sufficient detail [4,5]. After the four mandatory fields stated by Panizzi [6], the structure of a bibliographic record grew up to, for example, that of Scopus' 31 fields, where the most meaningful for the present analysis, apart from Title, Abstract and Keywords, are the fields Author, Affiliation and Nationality. The same fields can be extracted from the Clarivate Web of Science (WoS) database.

Scopus and WoS were used for the present research, because they are two databases that do not completely overlap, characterised by their relative strength in terms of information given by bibliographic records, and giving large thematic coverage of published documents. The growing Google Scholar free retrieval service for scholarly literature possibly gives more indexed documents, but they are accompanied by bibliographic records holding limited information, not suitable for the present research (see [7] for a critical review).

The bibliometric analysis of a set of bibliographic records helps to highlight thematic features characterising the published literature on the basis of words co-occurring in the documents as well as the presence of a social network of scholars focused on volunteered geographic information through document co-authoring [7] (see also [8,9]).

The bibliographic dataset analysed in the present research was made with bibliographic records from the Scopus (www.scopus.com) and WoS (www.webofscience.com) bibliographic databases by extracting the records on the basis of the Boolean expression "OpenStreetMap OR Open Street Map" included in the fields "Title OR Abstract OR Keywords", published up to and including December 2021.

The bibliographic dataset was analysed by the R library "bibliometrix" [7,10]. For each document, the full bibliographic record was downloaded. Then, a double check for duplicates was made, first automatically by the convert2df function; then, the dataset was manually curated by comparing the documents' DOIs to find further duplicates.

The abstract of a document is a stand-alone, concise and essential version of the paper [8,11]. It contains words regarding the main theme as well as more general themes relevant to the published research. Given that the abstract is a sort of mini-paper, the co-occurrences of the words composing the abstract were used to describe the main themes of

the documents published on the basis of OpenStreetMap data and to analyse the network of co-words forming the conceptual structure of the dataset [7]. Particularly, attention was given to bibliometrix's capability to extract and analyse bigrams, i.e., a type of two-word phrase [12–14], that help to identify the most covered themes and to hypothesise the semantic context the themes are involved in.

The impact of OpenStreetMap within the academic publishing environment was studied on the basis of the social structure inherent in the documents' dataset, i.e., by the analysis of three different networks given by the co-occurrence in each document, either institutions, authors, or countries [7,15].

Single nodes in a network can be completely isolated/connected, and their in-between "position" gives low/high complexity to the network. This feature was evaluated on the basis of the average value of [16]: (i) the density, proportional to the number of connections (ranging from 0 to 1); (ii) transitivity, which gives the probability of a node to produce a cluster with neighbouring nodes; (iii) degree of centralisation, which is equal to 1 (centralised) when the network is shaped as a star, while it is 0 in the case of a completely decentralised network; (iv) average path, i.e., the average of the shortest number of steps between two nodes.

## 3. Results

The query of the online databases (see Methods) led to the compilation of the OpenStreetMap (OSM hereinafter) bibliographic dataset, made of the bibliographic metadata of 2337 documents, i.e., the product of the scholarly community involved in volunteered cartography supporting the OSM initiative. The first document was that of Coast [1], where he announced the start in 2004 of the Open Street Map initiative. In the following five years, fifteen documents were published (see Figure 1); then, production increased almost linearly up to the 334 documents published in 2019, reaching a cumulative number of 1757 documents published within fifteen years. The same trend was not confirmed after 2019.

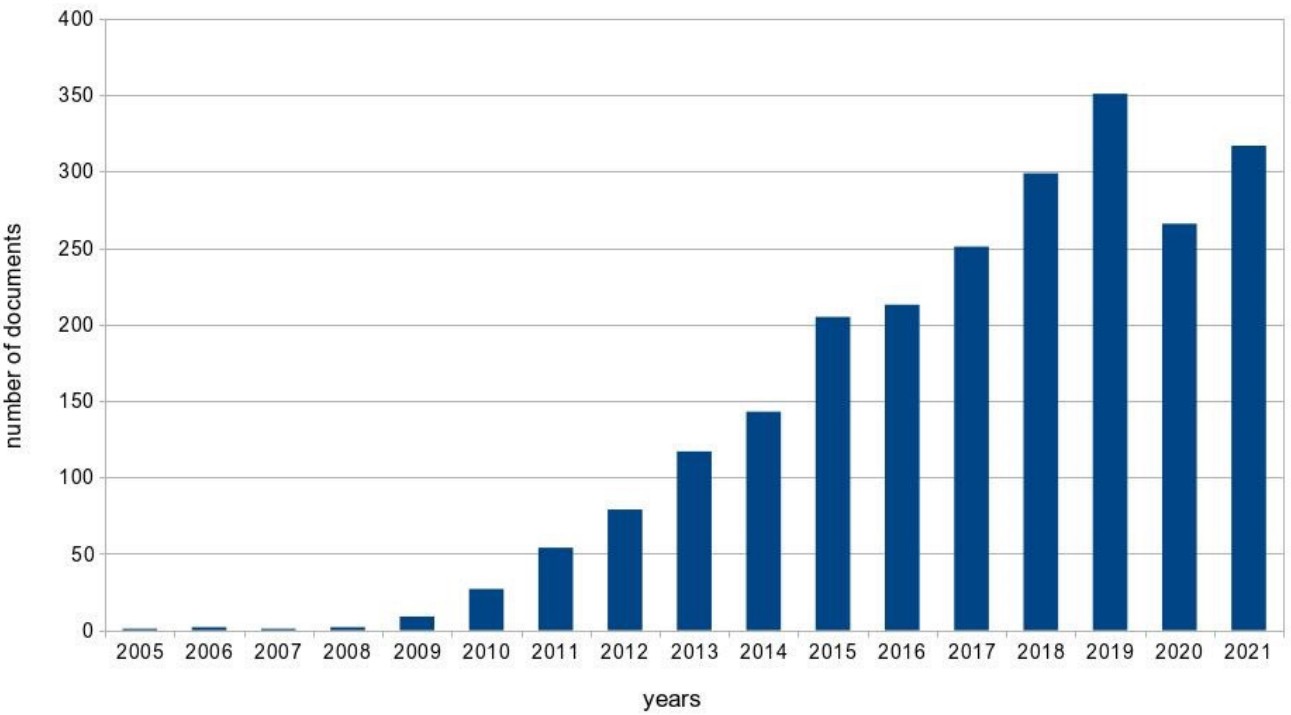

**Figure 1.** Number of documents per year.

The majority of documents (51%) were published (Table 1) in scientific journals, while less than half (46%) of the documents were published as conference papers or proceedings and 3% as book chapters or similar.

**Table 1.** Documents of the OSM dataset (total documents = 2337).

| Document Category | Document Type | abs. num. | Percentage |
|---|---|---|---|
| Journals | article | 1132 | 51 |
| | article in press | 5 | |
| | review | 31 | |
| | short survey | 5 | |
| Conferences | conference paper | 979 | 46 |
| | proceedings paper | 88 | |
| | conference review | 3 | |
| | article; proc. pap. | 1 | |
| Books | book chapter | 42 | 2 |
| | book review | 2 | |
| | book | 1 | |
| Other | correction | 3 | 0.6 |
| | data paper | 4 | |
| | editorial | 2 | |
| | erratum | 1 | |
| | note | 5 | |

abs. num. = Number of documents per type; percentage = percentage of documents per category.

More than five thousand authors (5419, see Table 2) contributed to publishing the OSM dataset; 187 of those (3.5%) produced 221 single-authored documents (9.6% of the dataset). Ninety percent of the dataset was composed of papers with an average of 3.6 co-authors per document, and with a collaboration index of 2.51 (i.e., the authors of the dataset documents divided by the number of multi-authored papers).

**Table 2.** Authors contributing to the OSM dataset.

| | |
|---|---|
| Authors | 5419 |
| Authors of single-authored documents | 187 |
| Authors of multi-authored documents | 5232 |
| Single-authored documents | 221 |
| Documents per author | 0.425 |
| Authors per document | 2.35 |
| Co-authors per documents | 3.59 |
| Collaboration index | 2.51 |

Collaboration index = authors of the documents divided by the number of multi-authored papers.

The conceptual structure of the OSM dataset was characterised by a total of 69,460 bigrams, 20 of those appearing very frequently (i.e., more than one hundred times in the dataset), followed by 725 bigrams appearing from ten to one hundred times (see Supplementary Materials Table S1). Only a few among them could be linked to the wide semantic context of geography (i.e., real places or objects in the world) or the natural environment, while the near totality of them can be referred to as informatics or cartography techniques.

Given the very large number of bigrams with low frequency, only those appearing at least ten times were analysed. The network produced by the bigrams' co-occurrence (Figure 2) showed a size of 745 nodes (Table 3), and even if edge density was low (0.12), the nodes were well connected with a degree centralisation of 0.71, and an average path between a pair of nodes of fewer than two steps (1.89).

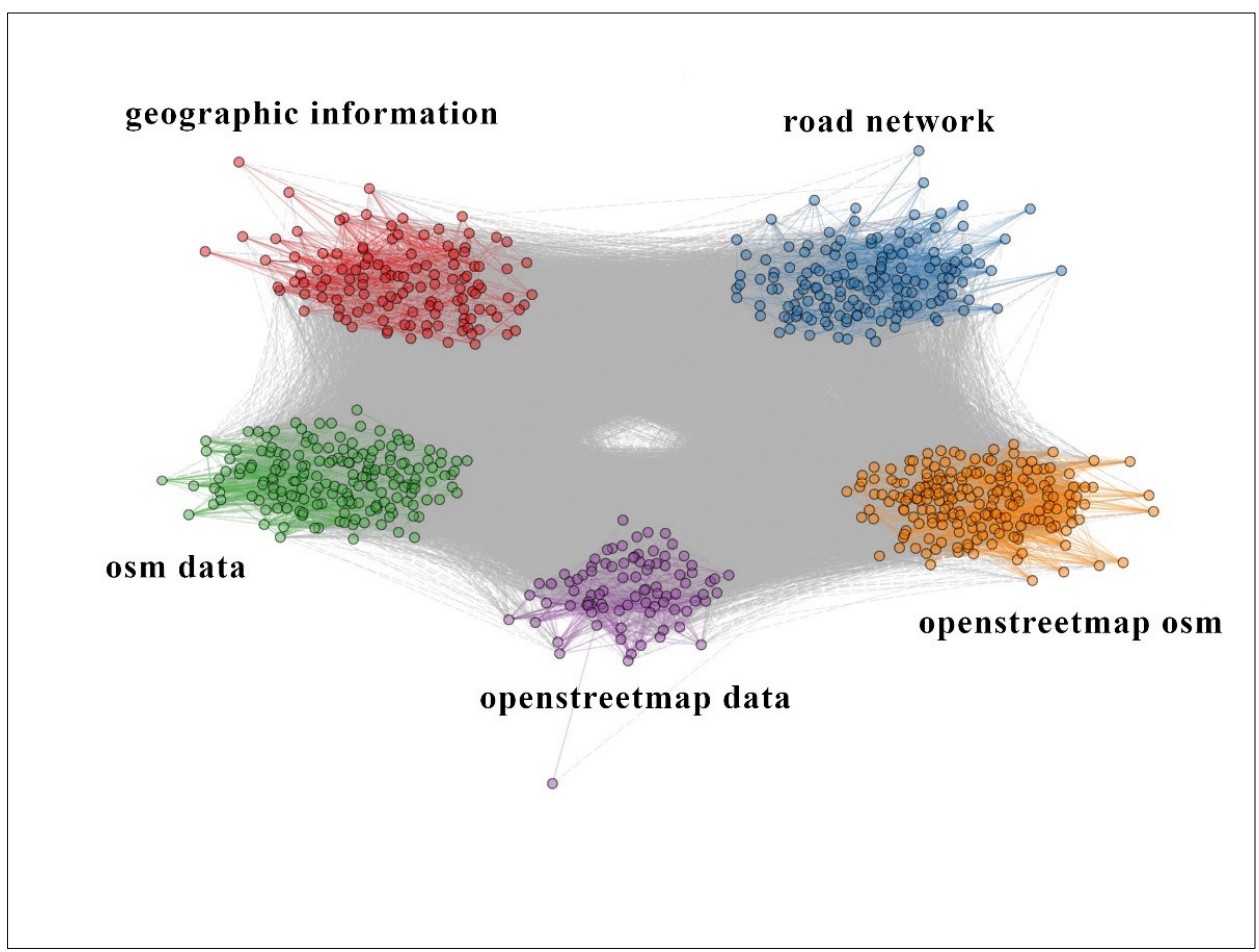

**Figure 2.** Co-occurrence network of the bigrams appearing at least ten times. The structure is characterised by five subnetworks: four with a larger size led by the bigrams "geographic information", "road network" "osm data" and "openstreetmap osm" and one smaller led by the bigram "openstreetmap data".

**Table 3.** Indexes of the conceptual network.

| | |
|---|---|
| Size | 745 |
| Density | 0.12 |
| Degree centralisation | 0.71 |
| Average path length | 1.89 |

The bigrams co-occurrence network showed a heterogeneous structure, organised into five subnetworks of bigrams, outlined in Figure 2 in different colours. All groups were characterised by sets of bigrams related mainly to data, mapping and geographic features, while bigrams related to true reality or ecosystems were very rare (see Supplementary Table S2).

The documents of the OSM dataset were published by 1779 institutions, but only two contributed to more than one hundred documents, i.e., Heidelberg University (183 documents) and Wuhan University (127), while 68 among the others contributed to more than ten documents, and more than a hundred institutions to between five and nine documents. Fewer than half of the institutions contributed to one document only.

Inherent in the OSM dataset, there is a network of relationships given by the collaboration among different institutions, i.e., a collaboration network (Figure 3). Under a general point of view, the collaboration network, among the institutes that published at least five documents (190), showed (Table 4) the lowest value of density (0.01), but the average path length and diameter were the widest within the social structure of the OSM

dataset (3.77 and 9, respectively). Nodes showed a low probability to make clusters, not far from 20% (transitivity 0.17).

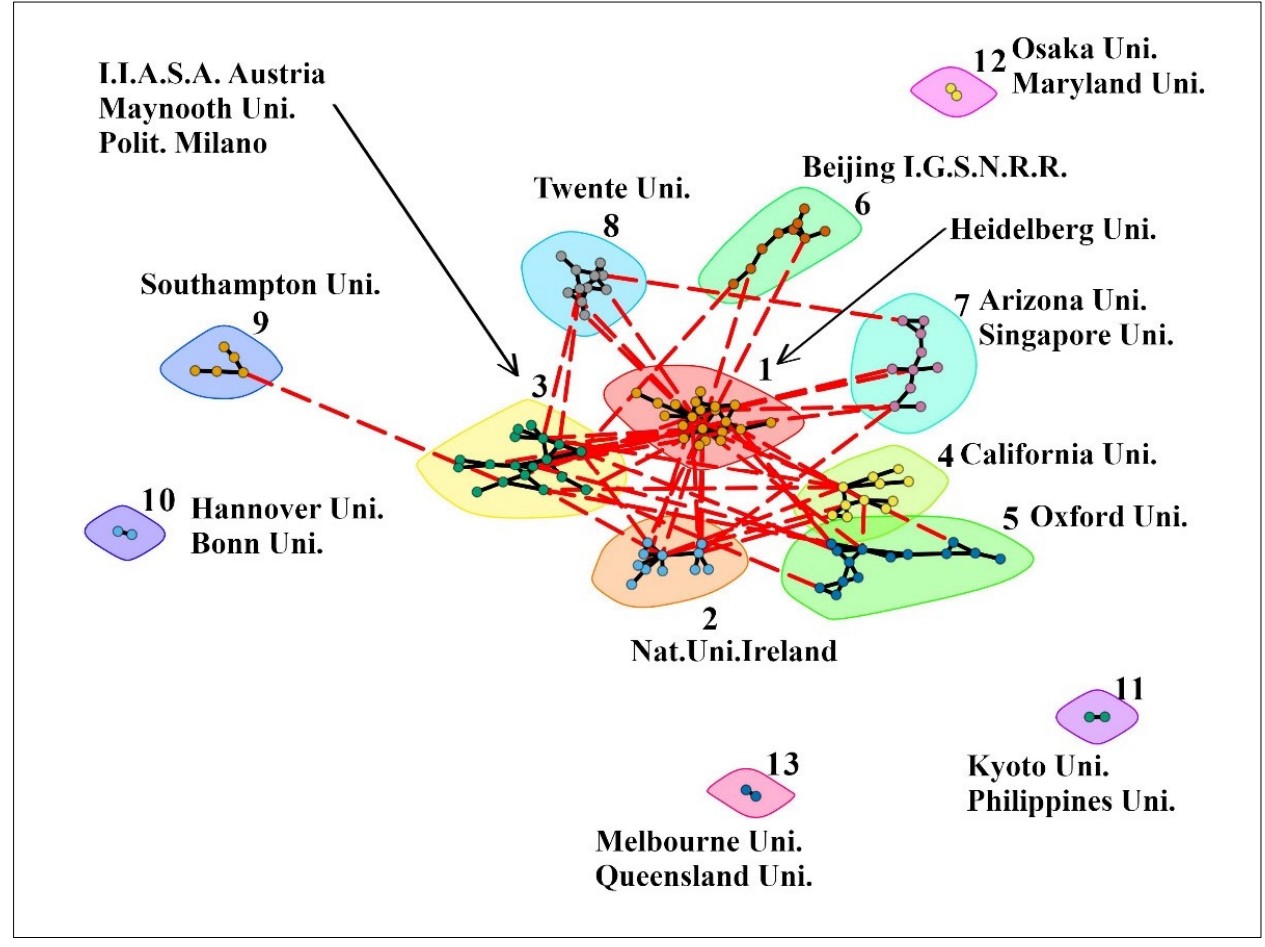

**Figure 3.** Collaboration network among the 190 most productive institutes, showing that there were eight large- to medium-sized and five small subnetworks. Clusters of institutes are highlighted by different coloured polygons. The title of the clusters is given by the institution with the higher degree of centrality (more than one in case of ex aequo). Within each cluster, there was no homogeneity on institute nationality. (1) Heidelberg University, Germany; (2) National University of Ireland; (3) Austria International Institute for Applied Systems Analysis; Maynooth University, Ireland; Politecnico di Milano, Italy; (4) California University, USA; (5) Oxford University, UK; (6) Beijing Institute of Geographic Sciences and Natural Resources Research, China; (7) Arizona State University, USA; National University of Singapore; (8) University of Twente, Netherlands; (9) Southampton University, UK; (10) Hannover Leibniz University, Germany; Bonn University, Germany; (11) Kyoto University, Japan; University of the Philippines; (12) Osaka University, Japan; Maryland University, Australia; (13) Melbourne University, Australia; Queensland University, Australia. See Supplementary Table S3 for a complete list of institutes.

**Table 4.** Indexes of the collaboration networks.

| Index | Institutes | Authors | Countries |
|---|---|---|---|
| Size | 190 | 201 | 89 |
| Density | 0.01 | 0.03 | 0.05 |
| Transitivity | 0.17 | 0.31 | 0.40 |
| Diameter | 9 | 8 | 4 |
| Degree centralisation | 0.14 | 0.02 | 0.34 |
| Average path length | 3.77 | 3.26 | 2.20 |

The network composed of the 190 most productive institutes was structured into thirteen clusters including 118 non-isolated institutes, with one cluster made of more than twenty institutes (see Supplementary Table S3).

More than five thousand (Table 2) authors contributed to the documents in the OSM dataset, and 201 among them contributed to five or more documents, while nearly four thousand contributed to one document.

The network of authors (Table 4) showed the lowest degree of centralisation (0.02). The probability of giving rise to clusters was beneath 50% (transitivity: 0.31). The network of the top 200 authors (Figure 4, Supplementary Materials Table S4) showed itself to be structured into 21 mainly small clusters grouping 185 authors, with five groups made of more than ten authors, and eleven groups with no more than two authors each.

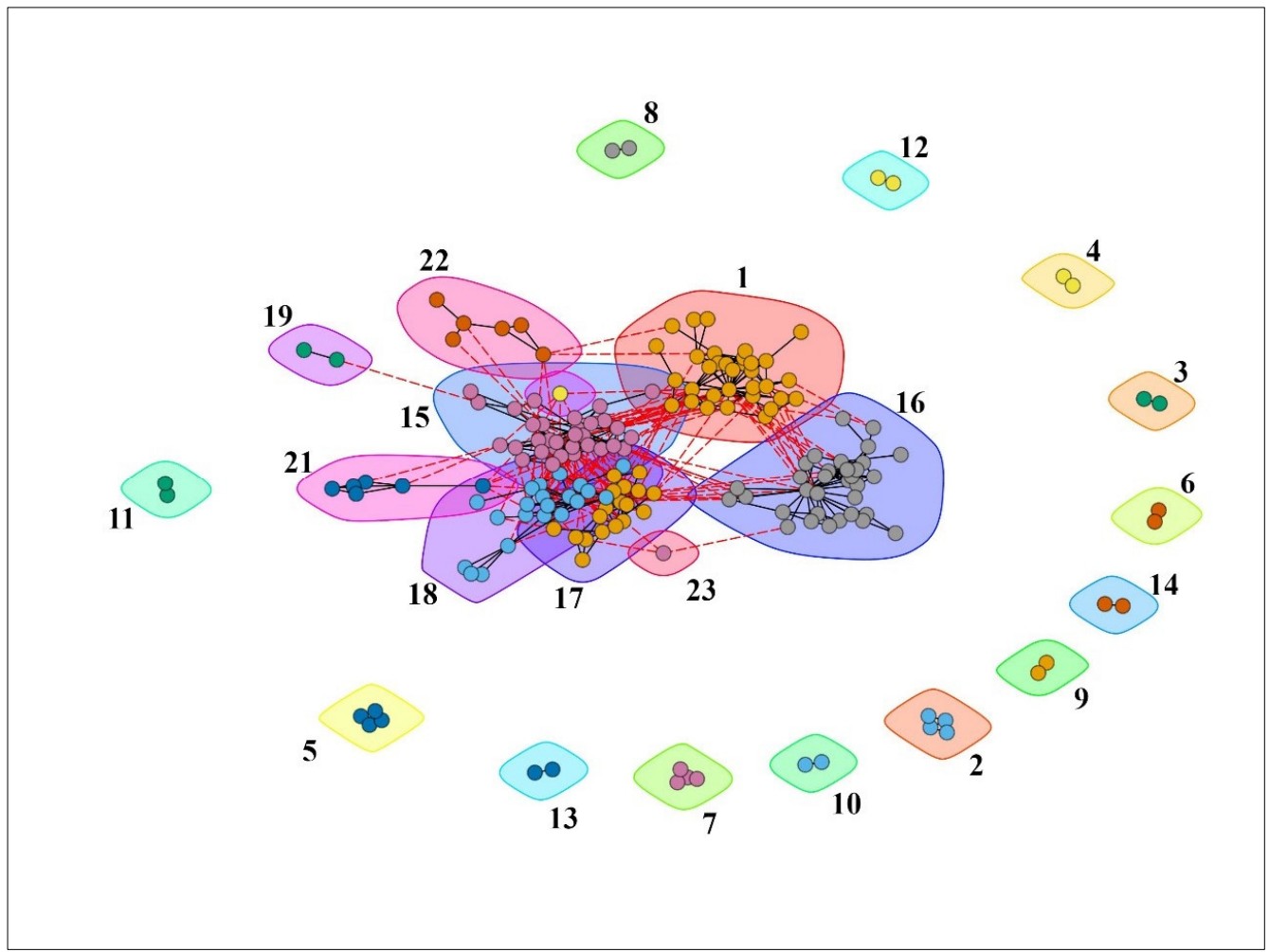

**Figure 4.** Collaboration network among the 201 most productive authors. The names of the authors were omitted, because the main focus is on the structure of the network, where it is possible to see a clump of clusters and several small "satellite" clusters. The clusters of authors are highlighted by different coloured polygons. See Supplementary Materials Table S4 for the authors' names.

The countries network was composed of few nodes (i.e., 89), with the highest indexes of density, transitivity and degree of centralisation (0.05, 0.4, and 0.34, respectively) within the social structure of the OSM dataset and the shortest average path. It was structured into three main clusters grouping 28 countries, being the remaining 61 isolated nodes (Figure 5, Supplementary Materials Table S5).

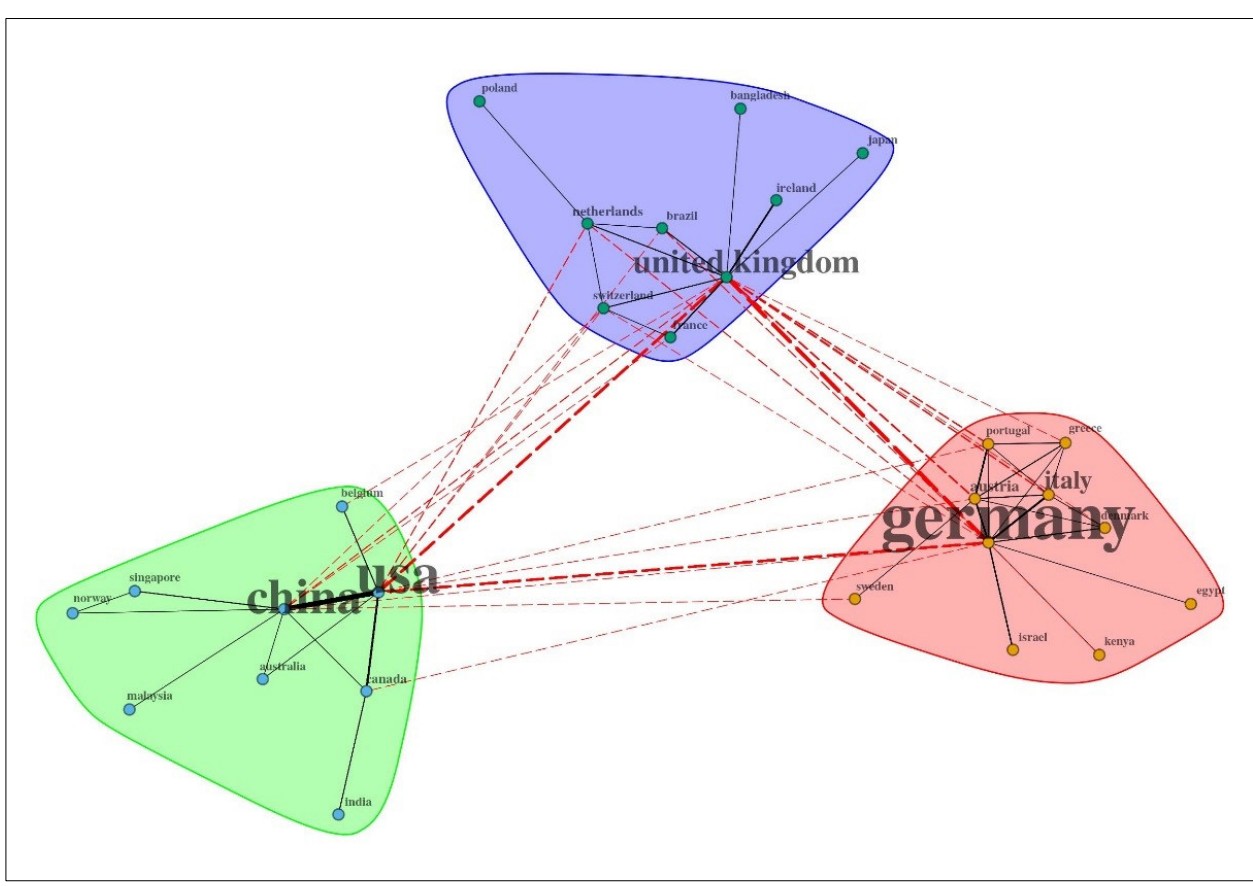

**Figure 5.** Collaboration network among all the non-isolated countries (88) involved in the production of the documents of the Open Street Map dataset. The leading countries are highlighted. A complete list is given in Table S5.

## 4. Discussion

The Open Street Map (OSM) initiative started as one of the many opportunities offered by the virtual space of the World Wide Web, and probably it was initially perceived simply as a starting novelty in computer science. It is likely that the scientific community did not notice or underestimated its potential, because the area of volunteered geographical information was not as consolidated at that time, as outlined by Goodchild [2], due to the fact that enabling technologies and OSM's popularity were both still emerging. This explains the slow scientific document production in the early years of OSM (Figure 1), when an average of three documents were published per year from 2004 to 2009. Then, documents were published with a more or less steady increase, possibly parallel to the distribution of the regions captured by OSM data (see also Supplementary Materials Figure S2).

This provides an answer to the starting point of the present research, i.e., the growing production of documents over time shows that OSM was not self-serving and devoted to the acquisition of pure geographical data, while it was accompanied by consistent document publishing over time.

The type of documents produced by volunteered cartography within OSM activity suggests that the immediacy of communicating new achievements was one of the main issues, because it was found that conference documents were nearly half of the total documents, possibly to speed up the dissemination of results that leveraged OSM, by virtue of the fast result-to-communication manner of conference environments. At the same time, the collaboration index (i.e., the authors of all documents divided by the number of multi-authored documents) suggests that such a need for dissemination was not an impulse for co-operation, because the whole set of authors did not contribute equally to global document production, given that the collaboration index was lower than the average

number of co-authors per document. Rather, as the analysis of the co-operation networks suggests, it is possible to hypothesise that authors, as a usual habit in scientific production, organised themselves into heterogeneous research groups focused on different research topics, where OSM was shared as a common platform.

The conceptual structure of the OSM dataset concentrated on terms of a predominantly technical nature, which can be integrated into a semantic context of a general nature and not primarily focused on one or more specific themes. Similar discussion is supported by the analysis of Yan et al. [17] (in relation to the wider context of volunteered geographic information), where they showed that great importance was given to topics related to data quality and credibility and to information handling and integration.

The technical nature of the terms was shown by the list of the most frequent bigrams, whereby the most frequent term was geographic information, relating to the semantic context of geography, followed mainly by terms that had no close relationships to real life (e.g., OSM data, spatial data and volunteered geography; see Supplementary Materials Table S1), even though, if not frequently, some terms linked to the real landscape did appear (e.g., road network, land cover and urban land); however, it seems this was more the exception than the rule. To obtain the first term linked to environmental issues, we needed to reach the 124th position, where the bigram "green spaces" was used 31 times. This was the first appearance of a term that was also used in the Corine Land Cover classification (see https://land.copernicus.eu), while the term "forest", one of the most ecologically sound, ranked 664 on the OSM bigram list (see Supplementary Materials Table S1). A similar list based on the terms written in the titles of the documents yielded the same results (not reported). Even if [17] found that volunteered cartography is a multi-perspective field leading to diverse research directions (e.g., social sciences and environmental classification), within the particular case of OSM, such a multi-perspective is shown to be more oriented to the technical aspects of data acquisition. This gives an answer to the question regarding preferred themes in the publications produced by the OSM initiative.

Due to the frequency of terms, the documents in the OSM dataset formed a co-occurrence network of terms (conceptual network) in which connections were made mainly to subjects not closely related to real-life geography, ecology and economics. This conceptual network proved stable, as the diversity of the 746 terms was organised into a network complexity with high degree of centralisation (=0.71), where the low edge density can be interpreted as a lack of redundancy; thus, although information flowed through a network that was not very branched, it could take advantage, however, of the shortest average path (1.89) between two words.

This means that at least the main terms of the OSM dataset were easily interconnected, i.e., most documents dealt with themes that were semantically interconnected. This can be interpreted as an initial step in the development phase of the conceptual network, where the outlines of new thematic subnets may be expected in the future. Some authors [17,18] suggested that one of the needs of OSM is a better and standardised means of data input to support data quality. While, if it is true, data quality will ensure the reliability of OSM data, it is likely that the development of more thematic subnets will enhance the information driven by the semantic structure of OSM data, where this could be achieved by giving more attention to issues related to real life, keeping a high level of data quality.

Is such a conceptual network the result of the co-operation between the actors (i.e., institutes, authors and countries) that maintain OSM? It was determined that the contribution of institutes was unevenly distributed, with only a few institutes contributing many documents, with document production concentrated in a few centres, mainly universities.

These institutes created a thin collaboration network with isolated nodes, where the clusters of institutes offered an overall situation of low connection (as illustrated in Figure 4 and Table 4), with long paths for information to follow to traverse the collaboration network. The fact that the groups inside the network were made of institutes from different nationalities suggests that co-operation was active within the groups of institutes and that it was transnational, while between the groups, co-operation was active to a lesser degree,

many of them being isolated from each other regardless of their size. It was found that the rank order of the institutes based on the centrality index was not reflected in the number of documents published by the institutes; therefore, the most productive institutes were not necessarily the most collaborative.

The same holds true for the network of authors, where most of them were clustered into 21 groups with many of them behaving like "satellites" made of a couple of authors. The network showed a collaboration environment fragmented mainly into small groups not connected to each other, apart from a small assemblage of five connected groups, making the actual network of authors.

Considering the authors' country as the node of the network, the same structure of the institutions seems reflected by the 28 connected countries, where high values of centralisation were likely due to the small size of the network. This can be interpreted as if information flowed between a limited number of countries without co-operating with most of the others. The main countries contributing to the collaboration network were Germany, USA, China and UK, showing a sort of European network skewness (see Figure 5). A similar network structure was suggested in [19] in the wider field of volunteered geographic information.

From a general point of view, the research themes developed by volunteered cartography within OSM showed that the application-oriented side of OSM is mainly generalist and partly divided into specific fields, since it would emerge from an eventual structure into subnetworks that were in fact partly recognised. The results may indicate that much attention has been paid to methods, sampling techniques and data entry, while applications for real-world cases or for verifying the reliability of OSM data applied to real-world cases have been less documented or somewhat neglected, at least as far as published documents are concerned. It is clear that when reading the results of the present study, it must be borne in mind that they are derived from a simple statistical analysis of bibliographic metadata, i.e., not from a critical review of the texts.

The main features of the conceptual structure indicated that the published documents did not deal in detail with geographical reality. This is probably an indication of the need to transfer the data and activities carried out by OSM to a more experimental scientific environment, where it may be important, for example, to evaluate the effectiveness of OSM's interaction with Humanitarian OSM as stressed by [20].

The social network was thin and slow, and this can be interpreted not as a consequence of a sort of protectionist attitude among institutes or scholars but probably as a consequence of the "technicality" characterising OSM, that is, because the documents deal mainly with technical aspects of data management. It appears that the major effort of scholars was directed at the best way to acquire, tag and render data. In this way, the knowledge acquired on the field finds it difficult to circulate and to be applied to solve real problems. It would perhaps be the opposite if many documents dealt with non-technical study cases.

It is reasonable to say that the sharing of OSM data could be improved by giving more attention than what is actually given to publishing papers on real-life cases, so that the documents published within, or as the output, of the OSM initiative could contribute to the production of a more heterogeneous conceptual structure, a better collaboration between scientists and institutes, and a dynamic intellectual structure in which concepts flow between all of the different actors in the publishing system. This would probably help to better standardise protocols for OSM application for the problem solving (rather than documenting) of real cases, such as those that occur with a certain frequency or severity such as floods, earthquakes and droughts. A similar discussion was developed in [17] that outlined the potential of volunteered geographic information in post-disaster and crisis recovery.

## 5. Conclusions

A new approach to studying the impact of OSM volunteered cartography was proposed on the basis of bibliometric analysis of the documents published within the initiative

of OSM. It was found that OSM volunteered cartography needs to be linked more to the publication of scientific papers where real geography is analysed and discussed in parallel with stressing data quality. It may be that the latter is quite easy to reach in the era of artificial intelligence, but case studies on real life need to take into account that they have to be driven by human volunteered geographers.

The OSM project, one of the main epiphenomena of volunteered cartography, has brought together scientists who have collaborated on the publication of several documents dealing with geographical data for updating the OSM database.

The research groups that created the conceptual network showed a common cultural background, which was mainly concerned with technical aspects of the management of cartographic data.

The research groups were not well networked within the social network, which means that the scientific production generated by the volunteered cartography has little impact on the exchange of information within the intellectual network. It suggests that possibly the academic publishing environment is not as collaborative as expected (and as discussed by some authors [21–23]).

The OSM database currently focuses mainly on Europe and parts of the USA (see https://osm-analytics.org), showing that there could be an inclination to concentrate knowledge, skill and technology within a few research groups with the risk that some part of the added value provided by OSM could be, in some way, monopolised. Therefore, it should be expected that OSM will become a more common tool in the scientific community based on geographical data (e.g., cartographers, ecologists and sociologists) by implementing world regional coverage of volunteered geographic data. This could be a great help in applying OSM data to a larger number of actual cases (e.g., the Humanitarian OpenStreetMap Team at https://tasks.hotosm.org/), where the aim will be to solve problems such as health or climatic emergencies or to highlight critical/endangered ecosystems at risk of biodiversity loss.

**Funding:** This research received no external funding.

**Supplementary Materials:** The following are available online at https://www.mdpi.com/article/10.3390/ijgi11070410/s1, Figure S1: Map of buildings data input over time; Figure S2: Chart of the number of buildings over time; Table S1: List of the bigrams extracted from the OSM dataset; Table S2: Clusters of the bigrams co-occurrence network; Table S3: Clusters of the institution collaboration network; Table S4: Clusters of the authors collaboration network; Table S5: Clusters of the countries collaboration network. Dataset used in this paper: OSM.RData.

**Institutional Review Board Statement:** Not applicable.

**Data Availability Statement:** The dataset is available as a supplementary material, formatted as R data (OSM.rdata).

**Conflicts of Interest:** The author declares no conflict of interest.

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
