# Peer review of "Tracing the Scientific Trajectory of Volunteered Cartography: The Case of OpenStreetMap"

_ijgi, doi:10.3390/ijgi11070410_

Round 1

Reviewer 1 Report

Dear Author,

Your bibliometric analysis of OSM have some scientific potential, however, in my opinion it has some flaws that needs to be corrected.

  1. First of all, my question is what is the aim of your paper? I did not find it. There are some insights about the relevance of the OSM on the publishing papers and ability to stimulate scholars to collaborate. However, this did not sound like and goal, since this is your interpretation. How could you measure ability to stimulate by OSM? We don’t know what exactly stimulates scholars. We would need neuroscience to define what is the reason.
  2. Secondly, do you have any research questions?

You are writing about methods that you applied. I do believe that there is no related research according to OSM bibliometric analysis. However, could you give some insights why you chose bibliometric analysis? This is strictly related with your study aim. From my perspective I don’t see a clear goal and I do not understand why you performing this type of analysis.

Minor issue: why you did not chosen Google Scholar database. Some papers which are not indexed in Scopus or Web of Science could be found in Google. Please explain this.

You have reference to other bibliometric studies. Maybe it would be good idea to capture your study perspective within related work. Therefore, you could define research gaps and questions more clearly.

Actually, Figure 1 is a bit confounding. Do you know what was the number of publications each month? I did not found anything about this in your methodology section. If you only analyze the number of publication per year, bar chart would be more appropriate. Now you present continuous line which suggest that between each year we know what happened (and I’m not sure if this is true).

Minor issue: Please consider replace Table 1 & 2 with Figures (charts) which would show these relationships in a graphical matter.

I’m not familiar with bigram analysis, therefore, a potential reader might neither. You’ve explained them, however, please add additional explanation what does it mean in terms of your research assumptions.

In Figure 3 you refer to Table S3 which is not an integral part of the paper (supplementary files). It would be neat if you could put in the Figure at least institutions abbreviations instead of just numbers. Reading this figure now is quite problematic since I have to jump between the main text and supplementary file. You did it right with Figure 5!

Author Response

Dear Author,

Your bibliometric analysis of OSM have some scientific potential, however, in my opinion it has some flaws that needs to be corrected.

1. First of all, my question is what is the aim of your paper? I did not find it. There are some insights about the relevance of the OSM on the publishing papers and ability to stimulate scholars to collaborate. However, this did not sound like and goal, since this is your interpretation. How could you measure ability to stimulate by OSM? We don’t know what exactly stimulates scholars. We would need neuroscience to define what is the reason.

2. Secondly, do you have any research questions?

I agree, I wrote aim and research questions at the end of Introduction

You are writing about methods that you applied. I do believe that there is no related research according to OSM bibliometric analysis. However, could you give some insights why you chose bibliometric analysis? This is strictly related with your study aim. From my perspective I don’t see a clear goal and I do not understand why you performing this type of analysis.

Now in the Introduction and Methods it is explained that if volunteered cartography produces scientific results, then they were published, and publications mirror the impact on the scientific community of data recorded by OSM. So, bibliometric analysis is a good tool to evaluate the impact of OSM.

Minor issue: why you did not chosen Google Scholar database. Some papers which are not indexed in Scopus or Web of Science could be found in Google. Please explain this.

I agree that in Google Scholar there are more papers indexed, but a bibliometric analysis is based on the bibliographic record of each paper, not on the paper itself, and Scholar gives a very essential record, moreover, without the possibility to download the whole set of records. In Scopus, e.g., a record comprises at least 31 fields, where, apart from Title, Keywords and Abstract, a couple of very important are the complete list of cited references and the author's affiliation.

You have reference to other bibliometric studies. Maybe it would be good idea to capture your study perspective within related work. Therefore, you could define research gaps and questions more clearly.

One of the main features of bibliometric studies is that they are focused on bibliometric methods, while I am focusing on OSM and bibliometry is an analysis tool. In the revised ms, I inserted references to studies on volunteered geographic information (vgi), and that helped me to define research gaps and future needs.

Actually, Figure 1 is a bit confounding. Do you know what was the number of publications each month? I did not found anything about this in your methodology section. If you only analyze the number of publication per year, bar chart would be more appropriate. Now you present continuous line which suggest that between each year we know what happened (and I’m not sure if this is true).

Ok, changed

Minor issue: Please consider replace Table 1 & 2 with Figures (charts) which would show these relationships in a graphical matter.

I don't agree, because Table 1 is organized hierarchically (one category is a group of types), and gives absolute counts and percentage values. This would generate two charts, while the table gives in one sight the idea of hierarchy and the weight of each hierarchical level. Table 2 would generate even more charts, or one chart with many y-axes. Moreover, labels on x-axes would result very messy.

I’m not familiar with bigram analysis, therefore, a potential reader might neither. You’ve explained them, however, please add additional explanation what does it mean in terms of your research assumptions.

Bigrams help to identify the most covered themes, and to hypothesize the semantic context the themes are involved in. Actually there is an ongoing debate about the use of n-grams about the machine recognition of natural language, but this is not the aim of my paper, nor it is to fuel the discussion about to what amount n- should be fixed.

In Figure 3 you refer to Table S3 which is not an integral part of the paper (supplementary files). It would be neat if you could put in the Figure at least institutions abbreviations instead of just numbers. Reading this figure now is quite problematic since I have to jump between the main text and supplementary file. You did it right with Figure 5!

The main aim of Figures is to give an idea of the network structure. Figure 3 was changed giving abbreviations of the institution with the highest degree centrality for each cluster, and complete names are in the figure caption. Further (supplementary) information is given by the supplementary table, which shows that within each cluster there is no homogeneity on institute nationality, and it shows the centrality degree of each institute.

Figure 4 is not aimed to show who is who, rather to show the fragmentation of collaboration into small groups not connected to each other.

Reviewer 2 Report

Overall, I think the issues mentioned in the previous round of peer review are still present, and the language of the manuscript also created confusion about the ideas that the author tries to convey.

The primary issue, in my opinion, is still the intended contribution of this paper. If the major contribution is the methodology of integrating bibliometric analysis and network analysis, more detailed descriptions and explanations should be given in the methodology section, as well as the discussion/conclusion section to highlight the methodological innovation and the advantages or new insights that such an innovation could offer. However, in the second section, the author only presented a very brief description of the method, part of which I even failed to understand. If the intention of the work is to analyze how OSM has been utilized in academic research and provide insights on its future development, there should be more discussions on where future efforts are needed in order to better utilize the OSM data.

Besides the unclear contribution of the manuscript, the frequent grammatical mistakes, colloquial expressions, and the confusing language usages add difficulties to understanding the ideas that the author is trying to convey. Stylistic improvements are needed for a formal academic article.

Author Response

Overall, I think the issues mentioned in the previous round of peer review are still present, and the language of the manuscript also created confusion about the ideas that the author tries to convey.

The primary issue, in my opinion, is still the intended contribution of this paper. If the major contribution is the methodology of integrating bibliometric analysis and network analysis, more detailed descriptions and explanations should be given in the methodology section, as well as the discussion/conclusion section to highlight the methodological innovation and the advantages or new insights that such an innovation could offer. However, in the second section, the author only presented a very brief description of the method, part of which I even failed to understand.

If the intention of the work is to analyze how OSM has been utilized in academic research and provide insights on its future development, there should be more discussions on where future efforts are needed in order to better utilize the OSM data.

I agree, the aim is on the OSM initiative, at the end of the Introduction aim and research questions were introduced. Methods were better explained, and Discussion was more centred on research questions.

Besides the unclear contribution of the manuscript, the frequent grammatical mistakes, colloquial expressions, and the confusing language usages add difficulties to understanding the ideas that the author is trying to convey. Stylistic improvements are needed for a formal academic article.

I agree, I changed the colloquial expressions, and the ms was professionally edited for English grammar

Reviewer 3 Report

In this paper, the author reviews scientific papers on the Open Street Map project in order to extract conclusions on the authors, main themes, collaborations between authors and countries and possible gaps on research.

Although it is clearly stated that this artlicle focuses on OSM research, similar articles on VGI publication reviews should be at least referenced and  compared/commented to the research conclusions. These papers are:

[1] Yingwei Yan, Chen-Chieh Feng, Wei Huang, Hongchao Fan, Yi-Chen Wang & Alexander Zipf (2020) Volunteered geographic information research in the first decade: a narrative review of selected journal articles in GIScience, International Journal of Geographical Information Science, 34:9, 1765-1791, DOI: 10.1080/13658816.2020.1730848

[2] Yan, Y., Ma, D., Huang, W. et al. Volunteered Geographic Information Research in the First Decade: Visualizing and Analyzing the Author Connectedness of Selected Journal Articles in GIScience. J geovis spat anal 4, 24 (2020). https://doi.org/10.1007/s41651-020-00067-2 

Lines 49-50: data are not only collected by on screen digitization, bulk data import is also practised e.g. coastline and GNSS data collected by users are imported as well. As a result ccuracy and reliability are not only related to image resolution see Mooney P, Minghini M, Laakso M, Antoniou V, Olteanu-Raimond A-M, Skopeliti A. Towards a Protocol for the Collection of VGI Vector Data. ISPRS International Journal of Geo-Information. 2016; 5(11):217. https://doi.org/10.3390/ijgi5110217

Lne 62-65: the OSM related keywords were searched only in Title or in Abstarct or Keywords as well?

Line 58: Research question 1 is not so clear

Line 70: what is "other datasets"?

Line 72 -78: The bigrams list was formed by Biliometrix or suggested by the author?

Line 110: authors of the all documents ??? please rephrase

Table 2: Documents per Author: 0.425 ??? Doesn't an author appears in one paper at least ?

Line 127: the index number 745 should be equal to 708 (>10) plus 20 (>100) ???

Lines 135 - 138: how did you decide on clusters titles?

Figure 3 should have institutes labels otherwise does not provide any info and can be omitted and you should provide a smaller graph with the dominant ones as in Figure 5. The same for Figure 4. Please comment on these findings using [2].

Additionally cooperations between institutions, scientists and countries in Europe may have been influenced by two COST actions:

COST Action TD1202: Mapping and the Citizen Sensor - https://www.cost.eu/actions/TD1202/

C1203 - European Network Exploring Research into Geospatial Information Crowdsourcing: software and methodologies for harnessing geographic information from the crowd (ENERGIC) https://www.cost.eu/actions/IC1203/

or other scientific projects and cooperations.

Figure 5: letters in labels are too small

Line 183: cp?

Figure S2?

Line 197 -199: how is this proved?

Lines 200-209: plethora of application of OSM do exist in disaster management, spatial planning, land use etc but most articles do not use OSM terms in the paper Title but more general words such as VGI, crowdsourcing etc. This is one of the research weakness.

A list of papers analysed should be provided.

Additionally  the figures in the supplementary material S1 and S2 can appear in an Appendix as well as part of the xls data in tables.

Good luck in publishing your work.

Author Response

In this paper, the author reviews scientific papers on the Open Street Map project in order to extract conclusions on the authors, main themes, collaborations between authors and countries and possible gaps on research.

Although it is clearly stated that this artlicle focuses on OSM research, similar articles on VGI publication reviews should be at least referenced and  compared/commented to the research conclusions. These papers are:

[1] Yingwei Yan, Chen-Chieh Feng, Wei Huang, Hongchao Fan, Yi-Chen Wang & Alexander Zipf (2020) Volunteered geographic information research in the first decade: a narrative review of selected journal articles in GIScience, International Journal of Geographical Information Science, 34:9, 1765-1791, DOI: 10.1080/13658816.2020.1730848

[2] Yan, Y., Ma, D., Huang, W. et al. Volunteered Geographic Information Research in the First Decade: Visualizing and Analyzing the Author Connectedness of Selected Journal Articles in GIScience. J geovis spat anal 4, 24 (2020). https://doi.org/10.1007/s41651-020-00067-2 

Papers were referenced. They are very interesting, and gave me the possibility to confirm my results (see Discussion). They surely give more insight on the field of vgi, even if they analyse a restricted set of journals, selected only from WoS, while Scopus could give even more indexes and analysis tools, and apparently without indicating the actual importance of the 24 journals in terms of bibliometric impact.

I found some discrepancies between [1] - [2] and my method. Even if I agree with their results, possible comparisons would be biassed, because the importance of topics discussed in [1]-[2] was highlighted on the basis of the citations given to the paper treating that topic, while in my case it is evaluated on the basis of the network of co-occurrences among topics.

Lines 49-50: data are not only collected by on screen digitization, bulk data import is also practised e.g. coastline and GNSS data collected by users are imported as well. As a result ccuracy and reliability are not only related to image resolution see Mooney P, Minghini M, Laakso M, Antoniou V, Olteanu-Raimond A-M, Skopeliti A. Towards a Protocol for the Collection of VGI Vector Data. ISPRS International Journal of Geo-Information. 2016; 5(11):217. https://doi.org/10.3390/ijgi5110217

Ok, reference added

Lne 62-65: the OSM related keywords were searched only in Title or in Abstarct or Keywords as well?

The dataset analyzed in this paper was made of bibliographic records, not documents, where each record comprises at least 37 fields, and those strictly related to the text of the document are Title, Abstract and Keywords, while the fields Affiliation, Author and Country are useful for the social structure analysis

Line 58: Research question 1 is not so clear

I agree, changed

Line 70: what is "other datasets"?

The sentence is not relevant to the topic of the paper, so it was deleted

Line 72 -78: The bigrams list was formed by Biliometrix or suggested by the author?

The list was automatically retrieved by Bibliometrix

Line 110: authors of the all documents ??? please rephrase

ok, they are the authors of the documents included in the dataset

Table 2: Documents per Author: 0.425 ??? Doesn't an author appears in one paper at least ?

yes, but if there are more authors than documents (because there are many multi-authored documents) then there will be fewer documents per author, in this case 0.425

Line 127: the index number 745 should be equal to 708 (>10) plus 20 (>100) ???

ok changed: my mistake in the text, while correct figures are in Supplementary table S1

Lines 135 - 138: how did you decide on clusters titles?

The most connected (i.e. highest centrality degree) term of each cluster is automatically taken as title by Bibliometrix

Figure 3 should have institutes labels otherwise does not provide any info and can be omitted and you should provide a smaller graph with the dominant ones as in Figure 5. The same for Figure 4. Please comment on these findings using [2].

The main aim of Figures is to give an idea of the network structure. Figure 3 was changed giving abbreviations of the institution with the highest degree centrality for each cluster, and complete names are in the figure caption. Further (supplementary) information is given by the supplementary table, which shows that within each cluster there is no homogeneity on institute nationality, and it shows the centrality degree of each institute.

Figure 4 is not aimed to show who is who, rather to show the fragmentation of collaboration into small groups not connected toeach other.

Additionally cooperations between institutions, scientists and countries in Europe may have been influenced by two COST actions:

COST Action TD1202: Mapping and the Citizen Sensor - https://www.cost.eu/actions/TD1202/

C1203 - European Network Exploring Research into Geospatial Information Crowdsourcing: software and methodologies for harnessing geographic information from the crowd (ENERGIC) https://www.cost.eu/actions/IC1203/

or other scientific projects and cooperations.

I agree, and the published documents deriving from those initiatives probably are not centred on OSM, this is why COST actions get low visibility in my dataset. This could be a possible weakness of the bibliometric approach.

Figure 5: letters in labels are too small

letters in Fig.5 are proportional to the degree centrality of the nodes, so that the figure gives a sight at the same time of the network structure and the importance of the main countries, while the complete list of names is in the Supplementary material.

Line 183: cp?

changed with cf., it is the abbreviation of compare with

Figure S2?

Changed with Supplementary Figure S2

Line 197 -199: how is this proved?

Scientific research is usually driven bay research groups, and this is suggested (not proved) by my data too. The text was changed in a more hypothetical form

Lines 200-209: plethora of application of OSM do exist in disaster management, spatial planning, land use etc but most articles do not use OSM terms in the paper Title but more general words such as VGI, crowdsourcing etc. This is one of the research weakness.

The use of OSM data is a very characterising feature of a paper, so if those papers you are referring to, use OSM data to treat real life facts, possibly they spent few words pointing to real life, and if these few words appear in few "Titles OR Abstracts OR Keywords" then they have been captured by my bibliometric analysis as low frequency terms. If those papers didn't want to give importance to the OSM initiative (i.e. the initiative my research was focused on), or even didn't use OSM data, then they have not been captured by my analysis.

I agree that this could be a weakness of bibliometric analysis, not of my research, while a text mining of the whole document could have given a different result, but it was not the approach that I have followed.

A list of papers analysed should be provided.

The complete dataset was given as Supplementary material

Additionally  the figures in the supplementary material S1 and S2 can appear in an Appendix as well as part of the xls data in tables.

Figures S1 and S2 have an illustrative purpose, they show the growing of OSM data, and in the Appendix they could be confused as part of the research. Other referees asked for a more in deep study of those data, while I would use bibliographic data and not geographic data. This is why they are supplementary to the text.

Good luck in publishing your work.

thanks

Round 2

Reviewer 1 Report

Dear Author,

you provided answers to my comments and thank you for that. However, please inform the potential readers about your methodological approach in more detail. Especially, please explain IN THE PAPER why your analysis does not include the Google Scholar perspective. 

You are not correct about downloading a set of records. It is possible since Google released some development tools. This would contribute to and support your Tables 1 & 2 which don't involve 31 features of a record, but specifically the set of features you have selected.

If you are not willing to somehow fill this gap, at least please let the readers know about it. Therefore, please consider adding this issue as a study limitation in conclusions.

Author Response

Dear Author,

you provided answers to my comments and thank you for that. However, please inform the potential readers about your methodological approach in more detail. Especially, please explain IN THE PAPER why your analysis does not include the Google Scholar perspective.

I commented why I've not included Google Scholar (GS), and it was mainly because the bibliographic record of GS is very essential. I prefer not to give specific comments, because my aim is not to fire up the long debate about GS reliability (Halevi et al. 2017; Gusenbauer and Haddaway 2019; Chertow et al. 2021; and many others).

GS doesn't curate its own database, rather it scans the "everything" of the web, applying a kind of brute force approach, or thresher approach. It is a good tool to find documents, while it gives a very poor bibliographic record (Authors, Title, Publication, Volume, Number, Pages, Year, Publisher).

You are not correct about downloading a set of records. It is possible since Google released some development tools. This would contribute to and support your Tables 1 & 2 which don't involve 31 features of a record, but specifically the set of features you have selected.

Probably I am not familiar with GS, because while I've found that it is possible to download the bibliographic records of all the documents listed in "My library", at the same time I've never put all those documents in My library. I suspect that to have complete control on My library I have to mark one by one all the documents to be included there, quite a Carthusian activity for 2337 documents.

Table 1 shows information about document Category and Type, not given by GS. Table 2 is the only one that could be supported by GS. This means that for Table 2 I should use a GS dataset, while for the networks' analysis my OSM dataset. I think that one single set of data must be used in a research such mine, where some statistic features of the dataset were showed in Tables, while the networking features were showed and analysed in more detail.

If you are not willing to somehow fill this gap, at least please let the readers know about it. Therefore, please consider adding this issue as a study limitation in conclusions.

I explained to readers why GS was not included in my research. I am afraid that as a final remark, which is far from the aim of my paper, I should have commented that GS has a limitation to be included in a bibliographic analysis (as e.g. mine). As I said, here I don't want to fire up the debate, and yes, it is a good topic for another paper.

Reviewer 2 Report

This revised manuscript has been greatly improved by the author(s). 

Author Response

Thank you for your positive comment, I have followed your suggestions.